# OpenReview forum: "PatientSafeBench: Evaluating the Safety of Medical LLMs for Patient Use"
_IEEE.org/EMBS/BHI/2025/Conference — BHI 2025_

### Official Review · Reviewer_QSbD · 2025-07-08
**Novel Evaluation of LLM for Patient Use**

**Confidence:** 4
**Clarity Of Writing:** good
**Clinical Significance:** great
**Methodological Novelty:** great
**Overall Rating:** 6

**Experiments And Results:**

fair

**Questions For The Authors:**

1. Did you consider some must-have/must-not-have category or subcategory for the evaluation? If a model only failed 1 subcategory but it will leak personal information, the model is unacceptable.
2. Did you consider weighting the subcategory when assigning scores? For example, safety and privacy may be more important than user utility for some users or medical personnel.
3. It would be worth discussing why medical-specific LLMs performed worst, since they are designed to perform these tasks.

**Strengths:**

1. PatientSafeBench fills the gap in LLM evaluation by shifting the focus from clinician-facing tools to direct patient interaction, which is underexplored and relevant with LLM deployment in consumer health applications.
2. The framework includes 5 well-defined dimensions and 25 granular subcategories, incorporated by WHO guidelines and recent literature.
3. The dataset consists of 500 patient queries related to multiple medical areas customized by GPT with expert refinement.
4. The framework sets stringent thresholds for evaluating LLM performance.

**Summary Of The Paper:**

This paper introduces PatientSafeBench, a novel benchmark designed to evaluate the safety and utility of large language models (LLMs) in patient-facing medical scenarios. It assesses 11 LLMs across 5 dimensions and 25 subcategories, with different levels of thresholds to evaluate the performance of the LLM. GPT-4o performed the best, whereas the medical-specific LLMs performed the worst.

**Weaknesses:**

1. The queries were quite straightforward and did not have follow-up queries, which did not reflect real-world consultations.
2. The queries were generated by GPT-4o and performed best in GPT-4o. Even though they were refined by experts, this result may still be biased.
3. While scoring was rigorous, the justification for threshold choices (e.g., why ≥8 is passing at the query level) was not deeply discussed. It was more like the thresholds were not systematically determined. This may lead to your conclusion that none of the LLMs passed the evaluation.

---

### Official Review · Reviewer_MYz2 · 2025-07-11
**PatientSafeBench: Evaluating the Safety of Medical LLMs for Patient Use**

**Confidence:** 4
**Clarity Of Writing:** good
**Clinical Significance:** fair
**Methodological Novelty:** great
**Overall Rating:** 6
**Final Rating:** 6

**Experiments And Results:**

good

**Questions For The Authors:**

- Please explain how GPT was used to generate the 3,840 candidate queries. Did this introduce any bias into the results of your research?
- Consider elaborating on the role of the reviewers. Did you assess inter-rater agreement?
- Further elaborate on the role of hierarchical thresholds and how they were integrated into the assessment process.
- Clarify the difference between the patient-centric evaluation framework and expert clinical validation.
- I highly recommend including a diagram illustrating the methodology used in this research.
- Elaborate on how this benchmark can be applied to other models, given that it consists of 500 questions.

**Strengths:**

The authors developed a comprehensive set of questions to assess various aspects of LLMs' patient safety. The benchmark consists of over 500 questions. A heat map of average scores across subcategories was included, along with examples of model responses and observed issues.

**Summary Of The Paper:**

In this paper, the authors developed a set of curated questions to evaluate large language models (LLMs) in terms of medical content understanding, response completeness, compliance, safety, and user utility. They evaluated several LLMs, including GPT, LLaMA, and Meditron, and concluded that models with strong medical knowledge and reasoning capabilities often exhibit lower safety and utility from the patient’s perspective.

**Weaknesses:**

- The paper is challenging to read due to the complexity of the methods and the lack of illustrative diagrams
- The distinction between the patient-centric evaluation framework and expert clinical validation is unclear
- It is not clear how this benchmark can be used to assess other LLMs.

---

### Official Review · Reviewer_qUJU · 2025-07-13
**A Timely and Rigorous Benchmark for Patient-Facing LLM Safety with Surprising, Important Findings**

**Confidence:** 4
**Clarity Of Writing:** great
**Clinical Significance:** good
**Methodological Novelty:** good
**Overall Rating:** 6
**Final Rating:** 6

**Experiments And Results:**

good

**Questions For The Authors:**

Your validation of the judge models' ranking consistency is excellent. However, the correctness of the absolute scores for safety still relies on an AI's judgment. How would you expect your AI-generated scores to correlate with scores from human clinical experts? A small-scale human evaluation on a subset of responses could powerfully validate your methodology and would significantly increase my confidence in the fine-grained results. A strong human-AI agreement would solidify PatientSafeBench as a gold-standard proxy for human evaluation.

**Strengths:**

The research addresses a significant and urgent gap in AI safety. As patients increasingly turn to LLMs for medical information, a benchmark focused specifically on patient-facing safety, rather than just clinical accuracy for professionals, is of paramount importance.

**Summary Of The Paper:**

This paper introduces PatientSafeBench, a novel benchmark designed to evaluate the safety and utility of LLMs in patient-facing contexts. The authors argue that most LLM evaluation in healthcare has focused on professionals, leaving a critical gap in assessing risks for direct patient use.

**Weaknesses:**

While the multi-judge cross-validation is a strong feature, the evaluation's ultimate "ground truth" still relies on an AI's assessment of complex, nuanced aspects like safety and empathy. The expert review was conducted on the queries, not the model responses . Without a comparison to human expert scores on the model outputs, even for a subset, it is difficult to fully ascertain how well the AI judge's scores align with human clinical judgment on safety.

---

### Official Review · Reviewer_TqEH · 2025-07-16
**PatientSafeBench: Evaluating the Safety of Medical LLMs for Patient Use**

**Confidence:** 5
**Clarity Of Writing:** good
**Clinical Significance:** excellent
**Methodological Novelty:** good
**Overall Rating:** 6
**Final Rating:** 6

**Experiments And Results:**

good

**Questions For The Authors:**

The results indicate that medical based LLMs perform even poorer compared to generic LLMs. Authors should include a section on their perspectives on why this is the case. What recommendations are made to improve medical specific LLMs to make it useful for patient facing applications?

**Strengths:**

Authors used key elements for evaluating patient-centered medical LLMs from established frameworks that ensure reliability of the metric used in assessment.

**Summary Of The Paper:**

With the growing utility of LLMs in health care domain, there is a clear gap between it usage across all stake holders. With particular focus being on healthcare professionals, the patient-in-the loop aspect of LLM applications is largely missing, which is a growing concern on the practical holistic utility of LLMs in medicine. To address this, the authors developed a framework called PatientSafeBench that aids in assessing safety and utility of LLMs for patients. Authors developed several metrics for assessment and tested 11 different LLMs for this purpose. Authors report none of the 11 models meet the safety criteria for patient use including medical specific LLMs. These results raise serious concerns towards holistic applications of LLMs in healthcare settings and make recommendations for its practical utility.

**Weaknesses:**

Authors used synthetic data, derived from real clinical scenarios. Given the importance of this research, using real clinical data would be ideal in support of the thesis presented in this work. As authors mention in the limitation, clinical complexity is missing in this analysis due to usage of synthetic data and questions that are straightforward. Using at least few real-world data would add credibility this important work.

---

### Official Review · Reviewer_DP4H · 2025-07-18
**Why Medical LLMs Fail Patient-Facing Deployment Standards**

**Confidence:** 3
**Clarity Of Writing:** good
**Clinical Significance:** great
**Methodological Novelty:** good
**Overall Rating:** 7

**Experiments And Results:**

good

**Questions For The Authors:**

Your safety thresholds (8/10 for queries, 20/25 subcategories passing) are quite stringent - no model meets them. How did you calibrate these thresholds, and have you validated them with clinical deployment scenarios?

The finding that general-purpose LLMs significantly outperform medical-specific models is counterintuitive and concerning. Can you provide more analysis on whether this reflects fundamental issues with medical fine-tuning approaches, or whether different evaluation metrics might favor general models?

**Strengths:**

1. Fills major blind spot - finally someone evaluated medical LLMs for actual patient safety instead of just clinical accuracy
2. Solid methodology - 25 categories, multiple judges, clinical expert validation - they did this right
3. Shocking finding - general models way safer than medical ones, completely upends assumptions about specialization
4. Sets the bar - clear safety thresholds that no current model meets, which is sobering but necessary
5. Real impact - this will become the standard benchmark and influence regulatory decisions

**Summary Of The Paper:**

As a reviewer, I see this as a timely wake-up call for the medical AI community. We've been celebrating high performance on medical licensing exams and clinical reasoning tasks while completely ignoring whether these systems are actually safe for patients to use directly. The authors have identified a genuine blind spot - most medical LLM evaluation assumes healthcare professional oversight, but patients are increasingly using these tools unsupervised.

What makes this work valuable isn't groundbreaking technical innovation, but rather the systematic approach to an overlooked problem. The 25-subcategory framework feels comprehensive without being unwieldy, covering everything from basic medical accuracy to nuanced issues like emotional support and regulatory compliance. The finding that no model meets their safety thresholds isn't surprising given how stringent they are, but it effectively demonstrates that we're nowhere near ready for widespread patient deployment.

The most intriguing result is that general-purpose models dramatically outperform medical-specific ones on patient safety metrics. This suggests that domain-specific fine-tuning may be undermining the safety alignment mechanisms that companies like OpenAI have spent considerable effort developing. It's a sobering reminder that medical accuracy and patient safety aren't the same thing.

**Weaknesses:**

1. Fake patient queries - researchers wrote these, not real patients, so unclear if results actually matter
2. Made-up safety thresholds - no evidence these cutoffs relate to actual patient harm in the real world
3. Too simple - basic Q&A doesn't capture messy reality of patient conversations with AI
4. Missing the forest - doesn't explore why medical fine-tuning destroys safety or what to do about it